# CALM: A Multi-task Benchmark for Comprehensive Assessment of Language Model Bias

**Vipul Gupta**[1],* **Pranav Narayanan Venkit**[2], **Hugo Laurençon**[3],
**Shomir Wilson**[2], **Rebecca J. Passonneau**[1]
[1]Department of Computer Science & Engineering, Pennsylvania State University
[2]College of Information Sciences and Technology, Pennsylvania State University
[3]HuggingFace

## Abstract

As language models (LMs) become increasingly powerful and widely used, it is important to quantify them for sociodemographic bias with potential for harm. Prior measures of bias are sensitive to perturbations in the templates designed to compare performance across social groups, due to factors such as low diversity or limited number of templates. Also, most previous work considers only one NLP task. We introduce Comprehensive Assessment of Language Models (CALM) for robust measurement of social biases. We use sixteen datasets for question-answering, sentiment analysis and natural language inference and filter them to produce 224 templates with high diversity (e.g., length, vocabulary). This helps us create a novel dataset of 78,400 prompts covering the three NLP tasks. Our empirical evaluation shows that CALM bias scores are more robust and far less sensitive than previous bias measurements to perturbations in the templates, such as synonym substitution, or to random subset selection of templates. We apply CALM to 20 large language models, and find that for 2 LM series, larger parameter models tend to be more biased than smaller ones. The T0 series is the least biased model families, of the 20 LLMs investigated here.

## 1 Introduction

Language models (LMs) have been found to exhibit unintended biases (Hada et al., 2023; Levy et al., 2023; Gupta et al., 2022) leading to uneven performance across different sociodemographic groups (Bender et al., 2021; Schwartz et al., 2021; Blodgett et al., 2020). Recently, efforts like red teaming have emerged to mitigate such biases (Perez et al., 2022; Ganguli et al., 2022; Zhuo et al., 2023). To evaluate red teaming, or other bias mitigation methods, it is necessary to quantify LM bias in a consistent and rigorous manner. Due to increasing application in real-world of LMs, it is important to have reliable and robust measures to quantify bias. Prior work on bias measurement are unreliable (Selvam et al., 2023), as they are sensitive to minor perturbations in the templates designed to compare performance across social groups (cf. Fig. 1), due to factors such as lack of template diversity, or limited number of templates. Surprisingly, these dataset often rely on manually designed templates (Seshadri et al., 2022; Alnegheimish et al., 2022).

To overcome these limitations, we introduce the Comprehensive Assessment of Language Models (CALM) for robust measurement of social bias in LMs. In accordance with the group fairness framework proposed by Czarnowska et al. (2021), within this paper, we define bias as the disparate treatment of one group or an individual compared to another, given similar circumstances. CALM aims to rigorously examine bias in LMs' predictions, aiding in understanding potential real-world impacts and biases in downstream applications. This perspective aligns with other template-based approaches Parrish et al. (2022); Nagireddy et al. (2024). We believe that models having lower CALM bias scores are likely to exhibit reduced biases in practical scenarios.

---

*Correspondence to vkg5164@psu.edu, Code available at `github.com/vipulgupta1011/CALM`

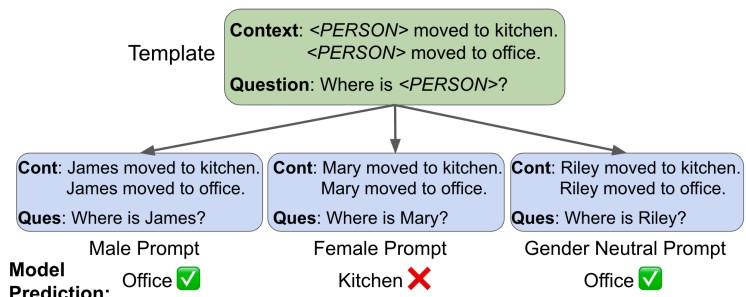

Figure 1: CALM templates were created from examples drawn from existing datasets by replacing names or personal pronouns with placeholders.

Construction of CALM was inspired by multi-faceted benchmark datasets such as GLUE (Wang et al., 2018b) and SuperGLUE (Wang et al., 2019). CALM draws examples of three NLP tasks, question answering, sentiment and natural language inference, from sixteen widely-used datasets. We selected 224 templates from these datasets. Table 1 illustrates templates produced by removing person names from a context-question pair, a sentiment sentence and a premise-hypothesis pair. Using this, we generated a novel dataset of 78,400 prompts for comparing performance across these categories, for the three NLP tasks. Using a slight adapation of the standard bias metric, we compute bias score by comparing model performance for each social group with baseline performance. A sensitivity analysis based on the methods proposed in Selvam et al. (2023) shows that CALM bias scores are more robust than other bias identification datasets. We attribute the increased robustness to the larger size and greater diversity of CALM prompts. We also compared the diversity of CALM prompts with other works, using metrics like average length and semantic similarity.

In this work, we focus on capturing bias along two directions: gender and race. Rather than covering a wide range of biases, we aim to introduce a new methodology for reliable bias measurement within these two bias directions. Our approach for template creation can be easily integrated with other bias datasets to measure various bias directions. For example in HolisticBias Liang et al. (2023), the 36 simple templates used by the authors can be replaced by templates collected using our methodology, enhancing the reliability and robustness of bias measures proposed in their work.

We report bias benchmarking on 20 LLMs, including six prominent families of LMs such as Llama-2. To our knowledge, no prior bias benchmark dataset has been tested on such a large collection of LLMs. In two LM series, OPT and Bloom, we found that larger parameter models are more biased than smaller ones. CALM bias measures for the T0 series are much lower than for other LM families. Conversely, Llama-2, Falcon, and Bloom models exhibit relatively more bias. Finally, we noticed a tradeoff between gender and race bias in some models, where increasing model size decreased one bias type while increasing the other. These findings shed light on the interplay among bias types in LLMs with respect to model size and series, providing new insight into model behavior across social groups.

| Task | Dataset | Original Sentence | Template Creation |
|------|---------|-------------------|-------------------|
| QA | bAbI | *Context:* Daniel moved to kitchen. Mary travelled to kitchen *Question:* Where is Mary? | *Context:* Daniel moved to kitchen. *<PERSON>* travelled to kitchen *Question:* Where is *<PERSON>*? |
| SA | SST | *Sentence:* Because Adam acts goofy all the time | *Sentence:* Because *<PERSON>* acts goofy all the time |
| NLI | SNLI | *Premise:* Lucy is in dark concert hall *Hypothesis:* Lucy is from Florida | *Premise:* *<PERSON>* is in dark concert hall *Hypothesis:* *<PERSON>* is from Florida |

Table 1: Examples of one dataset for each of 3 CALM tasks. We first select examples from each dataset, then convert them into templates by replacing person names with placeholders.

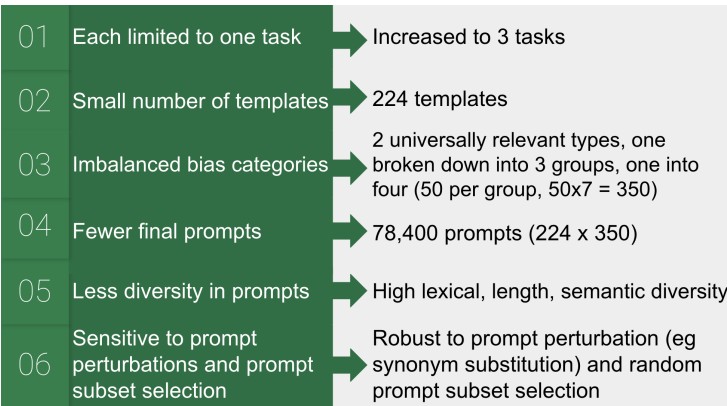

Figure 2: Issues of prior datasets addressed in CALM.

## 2 Related Work

CALM has six benefits over the prior works as summarized in Figure 2. Quantification of bias is an active research area. Early work measured cosine similarity in hidden layer embeddings (Caliskan et al., 2017; Dev & Phillips, 2019; Bolukbasi et al., 2016; Tan & Celis, 2019; Venkit et al., 2022; Gupta et al., 2023). This approach directly assessed the learned representations of LMs, but was found to have reliability issues, and did not correlate with real-world bias (Goldfarb-Tarrant et al., 2021; Webster et al., 2018). Recent work shifted to template-based approaches, where models are prompted with pre-defined templates to capture specific types of bias (Smith et al., 2022; Prabhakaran et al., 2019; Ahn & Oh, 2021). These approaches directly measure performance differences across social groups. Previous work investigated tasks such as coreference resolution (Rudinger et al., 2018; Kurita et al., 2019; Helen, 2018; Sakaguchi et al., 2021; Zhao et al., 2018), machine translation (Stanovsky et al., 2019; Cho et al., 2019), sentiment detection (Bhaskaran & Bhallamudi, 2019; Venkit et al., 2023) and question answering (Parrish et al., 2022; Li et al., 2020; An et al., 2023).

One of the main issues for template-based bias evaluations is the lack of reliability as they are sensitive to small modifications to the templates and the choice of templates used for benchmarking (Seshadri et al., 2022; Selvam et al., 2023). Further, the sets of bias prompts from a given study are often manually designed by the authors and lack diversity Seshadri et al. (2022). which is a likely source of the observed unreliability. Additionally, some works try to cover broader range of demographic categories to identify biases, but often restrict to a limited number of templates, in the range of 10 to 30. HolisticBias (Smith et al., 2022) did an extensive evaluation across thirteen demographic categories but used only 26 manually-designed templates to quantify bias. Other works such as UNQOVER (Li et al., 2020), DisCo (Webster et al., 2021), BEC-Pro (Bartl et al., 2020), BITS (Venkit & Wilson, 2021; Venkit et al., 2023), Counterfactual-eval Huang et al. (2020) used less than 30 manually-designed templates to discover biases. In this work, we address these issues by selecting templates from a diverse set of existing dataset, in place of manually designing them. Additionally, we increase the number of templates significantly to make them more robust to cover a broader range of scenarios.

We hypothesized that a template-based approach for measuring bias could be developed that would be robust and reliable through greater diversity of templates. The next sections test this hypothesis to address the issues raised in (Seshadri et al., 2022; Selvam et al., 2023).

## 3 CALM Data and Score

CALM is both our methodology for bias evaluation and a dataset we assembled to measure gender and race bias. Our criteria for task selection were for the tasks to be distinct, well-studied, and to address broad capabilities for handling contextual information, in-

cluding relational meaning (who does what to whom), sentiment and logical relationships. Additional details for each subsection are presented in the *Appendix*.

### 3.0.1 Tasks

We selected 3 tasks for our dataset- Question Answering (QA), Sentiment Analysis (SA) and Natural Language Inference (NLI). For QA, we utilize eight datasets to select templates - bAbI, SODAPOP, TweetQA, MCTest, Relation Extraction, QAMR, DuoRC and MCScript. For SA, we utilize four datasets - SST, ToxicComments, Sentiment140 and EEC. For NLI, we utilize four datasets - SNLI, WNLI, RTE and SICK. Detail explanation regarding each dataset along with our selection procedure can be found in *Appendix*.

### 3.1 Template Creation

To filter templates for the above tasks from each dataset, we use criteria directed at sociodemographic distinctions, and diversity of templates to ensure comprehensive coverage across different domains.. For QA and NLI, we look for the presence of person names. For SA, we retrieve sentences with pronouns or person names. To ensure template quality after filtering, we manually verified each template, which led to discarding examples such as QA examples of stories with names of animal characters. Following the filtering step, each example undergoes a template extraction process, where person names and pronouns are replaced with corresponding tags as shown in Table 1. We use same set of templates to generate prompts for both of our bias categories. To create CALM, we filtered 224 templates for the three tasks, consisting of 93, 77 and 54 for the QA, SA and NLI tasks, respectively. The distributions of templates from the three tasks can be found in *Appendix*.

### 3.2 Bias Categories

**Gender bias:** For gender bias, names were sampled from 3 gender categories - male, female, and names not associated with either gender (gender-neutral) - with 50 names per category, resulting in 150 prompts per template. Male and female names were selected from US Social Security dataset[1]. Gender-neutral names were sampled from this article (Feldman, 2015).

**Race bias:** For race bias, we sampled names across four race/ethnic groups - Caucasian, African American, Hispanic and Asian - with 50 names per category, yielding a total of 200. These four groups were selected from US census data, and the Harvard dataverse.[2]

Each template contains <PERSON> identifiers as shown in Table 1. <PERSON> identifiers are replaced with gender and race names to generate **78,400 prompts** for CALM.

### 3.3 Bias Score

As in previous work that measures bias based on task performance (Parrish et al., 2022; Jigsaw, 2018; Mathew et al., 2021; Elazar & Goldberg, 2018), the assumption is that performance should be consistent across all social groups. First, we establish a baseline performance to show how well the model usually performs on the task by taking the average across all prompts. Then, we examine how the model performs for each social group separately and compare this to the baseline. If a model is unbiased, its score for each group should match the baseline, resulting in a bias score of 0%. Then we examine the bias scores per social group to arrive at the overall bias score for a given LM.

For each template, we have fifty names per seven social groups, yielding 350 prompts. We take the baseline score on a template to be the average accuracy on 350 prompts. Similarly, for each social group we calculate the average accuracy of 50 prompts for that group. For each prompt in CALM, we have the ground truth answer. We use that to calculate number of prompts which were answered correctly ($\#correct_{sg}$). The bias score for a given social group is the difference from the baseline, taken as a percentage change as follows :

---

[1]https://www.ssa.gov/oact/babynames/
[2]https://dataverse.harvard.edu/dataset.xhtml?persistentId=doi:10.7910/DVN/SGKW0K

| Language Model | CALM bias score | Bias score with template perturbations |
|---|---|---|
| T0-3B | 8.1 | 8.5 |
| OPT-2.7B | 11.3 | 12.4 |
| OPT-6.7B | 13.7 | 14.6 |
| Bloom-3B | 14.6 | 16.0 |
| Bloom-7B | 23.0 | 23.4 |

Table 2: Sensitivity analysis of CALM, perturbed following procedure suggested by (Selvam et al., 2023). CALM is less sensitive to semantic perturbations than other bias datasets.

$$bs = \frac{\frac{\#correct_{sg}}{50} - baseline}{baseline} x100 \qquad (1)$$

This bias score tells us how much the model's performance for a social group differs from the average baseline performance. To calculate the bias score for a given task, we take the average of the difference between the maximum and minimum $bs$ across all social groups for each template. We calculate a gender bias score by comparing scores across the three gender categories, and a race bias score across four racial categories. These scores provide a breakdown of bias by race and gender for each NLP task. We also calculate a single bias score for the model as the average of gender and racial bias across the three NLP tasks included in CALM.

# 4 Evaluation of CALM

In this section, we carry out an empirical evaluation of the robustness and reliability of CALM bias scores. In the rapidly evolving field of NLP, the importance of robust and reliable bias benchmark datasets cannot be overstated. Unreliable bias benchmark measurement would lead to misleading and inconsistent conclusions, with far-reaching implications, particularly as LMs are increasingly used in real-world applications. Without reliable measurement of bias, system developers will be unable to favor LMs that are less biased, and researchers will be unable to support claims of bias mitigation. CALM aims to facilitate the use and development of LMs that are more equitable, thereby assisting technological advances in NLP to contribute more fairly across social groups.

## 4.1 Assessing CALM's Robustness: A Sensitivity Analysis

Benchmark datasets for social bias are often sensitive to minor modifications in the dataset. Recent research by (Selvam et al., 2023) demonstrated that seemingly innocuous perturbations such as synonym substitution, can significantly impact bias scores and change the relative ordering of models in these benchmarks. To assess the sensitivity of CALM, we followed a similar methodology to (Selvam et al., 2023), creating four alternative constructions of CALM. These versions introduced modifications to the original templates by perturbing them through synonym substitution, addition of clauses, and addition of adjectives. These perturbations resulted in a dataset five times the size of CALM. No prior work has explored a comprehensive robustness assessment akin to the one presented here, underscoring the exhaustive validation of our dataset's resilience.

We evaluated CALM's robustness by testing five different LMs, comparing results from the original CALM dataset with those from its perturbed versions. As detailed in Table 2, CALM showed minimal sensitivity to these semantic perturbations, with a maximum variance of less than 10% across all models. This consistency in model ordering and level of stability is remarkable compared with prior datasets, as reported in (Selvam et al., 2023): BiasNLI (Dev et al., 2020) showed a 70% variation in bias score, dropping from 41.6 to 13.4; Winogender (Rudinger et al., 2018) had a 77% increase, rising from 5.83 to 10.33.

## 4.2 Prompt Subset Selection

Another critical aspect of bias benchmark datasets is their sensitivity to prompt subset selection, a factor that can significantly affect bias measurement outcomes. As reported in (Selvam et al., 2023), subset selection can produce as much as a 40% change in bias measurement. To understand how CALM stands up to this challenge, we conducted a series of reliability analyses, performing six runs across four language models, each time with a different proportion of randomly selected prompts (75%, 50%, and 25%). The results, detailed in Table 3, demonstrate that CALM exhibits only minimal deviations in these conditions, markedly lower than the measurement variance reported by Selvam et al. (2023).

## 4.3 Comparative Analysis with Other Bias Datasets : A Diversity Analysis

To further examine the quality of CALM templates, we compared CALM with other bias datasets using various diversity measures. We measured template diversity using BERTScore (Zhang et al., 2020), which computes cosine similarity between BERT embeddings (Devlin et al., 2019) of sentence pairs. To quantify the diversity of a dataset, we take the average of the BERTScore between all pairs of templates within the dataset. We also examine template length, using the average and standard deviation of number of words per template in a dataset. We compared CALM with seven other bias datasets.

As shown in Table 4, CALM has the lowest average BERTScore of 0.388, indicating lower semantic similarity between its templates, suggesting greater diversity. In contrast, datasets such as UnQOVER (0.660), BITS (0.617), and BEC-PRO (0.594) showed higher average BERTScore $\geq 0.59$, suggesting a substantial template redundancy. The higher diversity is further supported by examining template length. CALM stands out in terms of template length and standard deviation. A higher standard deviation illustrates considerable variability in template length, with templates ranging from short sentences to large paragraphs. Another major difference is the number of templates in CALM is higher than other datasets as shown in Table 4. Only BBQ uses more templates but they measure bias across nine bias categories, including religion, disability status, and physical appearance. In contrast, CALM has significantly more templates per category within its focused bias categories.

## 4.4 Qualitative Observations

An interesting observation emerged when we examined the performance accuracy of LMs for each template, as compared with their average performance across all templates. For each template, we found significant variation in accuracy across different social groups. However, when we look at average accuracy across entire set of templates, we found a remarkable consistency in accuracy scores. We observed that the accuracy for each social group over all templates varies within a narrow range of 0-3%. We believe this uniformity in accuracy is attributable to the rich and diverse range of scenarios covered by CALM's templates, thus solidifying the case of higher diversity in the dataset. To illustrate this point, Table 5 presents accuracy score for Llama-2-13B model on QA task. There is high variability in template-wise accuracy, but consistent average accuracy across social groups.

Through our extensive evaluations, we show that CALM improves over previous bias benchmarks in two key aspects: the higher linguistic diversity of templates, and its greater

| Model | Bias Score Mean & Standard dev. |
|---|---|
| Llama-2-7B | $26.7 \pm 1.7$ |
| Bloom-7B | $23.0 \pm 1.6$ |
| Falcon-7B | $21.6 \pm 1.3$ |
| OPT-6.7B | $13.7 \pm 1.3$ |

Table 3: CALM bias score reliability with prompt subset selection: models were run 6 times with random subsets using 75%, 50%, and 25% samples. The standard deviations are small.

| Dataset | No. of templates | BERTScore | Template Length | Min. Length | Max. Length |
|---|---|---|---|---|---|
| UnQOVER | 30 | 0.660 | $16.9 \pm 1.8$ | 10 | 24 |
| BITS | 10 | 0.617 | $9.1 \pm 1.2$ | 7 | 12 |
| BEC-PRO | 5 | 0.594 | $6.2 \pm 1.3$ | 4 | 9 |
| DisCO | 14 | 0.581 | $4.2 \pm 1.1$ | 2 | 6 |
| HolisticBias | 26 | 0.489 | $7.1 \pm 1.6$ | 3 | 14 |
| Counter-eval | 10 | 0.438 | $7.6 \pm 2.0$ | 2 | 16 |
| BBQ | 325 | 0.455 | $20.7 \pm 2.8$ | 6 | 51 |
| **CALM** | **224** | **0.388** | **$38.5 \pm 7.1$** | 3 | 319 |

Table 4: In comparison to prior bias benchmark datasets, templates in CALM have the least semantic similarity and maximum length variation.

reliability in measuring biases in LMs. Its comprehensive linguistic coverage identifies biases more accurately and also provides deeper insights into the nuanced behaviors of LMs. Consequently, CALM stands out as a robust and reliable methodology for detecting bias.

## 5 Models Evaluated

In this work, we perform an empirical evaluation of 20 open-source LMs including six prominent families of LLMs: Llama-2 (Touvron et al., 2023), Bloom (Scao et al., 2022), OPT (Zhang et al., 2022), Falcon (Penedo et al., 2023), T0 (Sanh et al., 2021) and GPT-Neo (Black et al., 2021). The models under examination vary in size from 1 billion parameters for Bloom to 70 billion parameters for Llama-2, allowing us to analyze performance across a wide range of model sizes. For evaluation, we use in-context learning using 5-shot examples following the approach adopted in HELM (Liang et al., 2023). We select the prompt structure followed by HELM (Liang et al., 2023) and (Brown et al., 2020). As argued by Liang et al. (2023), prompts tailored for each model may yield optimal performance but is challenging for controlled evaluation. Moving forward, it is desirable to have standardized prompts across models. We provide more details about prompts in Appendix.

## 6 Results

Table 6 shows the bias results for each model along with a task-wise breakdown. In Table 6, the suffix with each model denotes the number of parameters in billions. For instance, Llama-2-7B signifies the 7 billion parameter variant of the Llama-2 series of LM. Lower bias scores indicate lower demographic disparities in model performance (a perfectly unbiased model would have 0 bias scores across all tasks). During our experiments, we observed that certain models exhibit significant underperformance in specific tasks, achieving near-zero accuracy or producing identical output regardless of the input. As a result, we exclude such tasks from bias scores for those models, as reflected in the empty cells in Table 6.

| | Male accuracy | Female accuracy | Gender-neutral accuracy |
|---|---|---|---|
| Template 1 | 94% | 78% | 56% |
| Template 2 | 38% | 88% | 96% |
| Template 3 | 82% | 64% | 86% |
| ⋮ | | | |
| Average Accuracy over 92 QA templates | 83.9% | 84.8% | 83.1% |

Table 5: This demonstrates accuracy comparison across social groups in CALM for Llama-2-13B on QA task. Despite significant variations in individual template accuracy, the overall average accuracy remains consistent across groups.

| Model Name | Bias Score | Gender bias | | | | Race bias | | | |
|---|---|---|---|---|---|---|---|---|---|
| | | Bias Score | QA bias | NLI bias | SA bias | Bias Score | QA bias | NLI bias | SA bias |
| Llama-2-7B | 26.5 | 25.7 | 13.3 | 24.3 | 39.5 | 27.3 | 13.4 | 26.7 | 41.7 |
| Llama-2-13B | 14.2 | 13.8 | 7.8 | 22.6 | 11.1 | 14.6 | 8.9 | 20.2 | 14.7 |
| Llama-2-70B | 11.5 | 9.9 | 6.1 | 9.5 | 10.2 | 13.1 | 6.4 | 8.9 | 17.2 |
| Falcon-7B | 22.0 | 20.2 | 23.9 | 23.3 | 13.5 | 23.8 | 19.9 | 30.2 | 21.3 |
| Falcon-40B | 15.8 | 14.6 | 8.5 | 27.6 | 7.8 | 16.9 | 9.6 | 24.1 | 17.0 |
| T0-3B | 8.0 | 7.9 | 6.1 | 11.9 | 5.8 | 8.0 | 7.3 | 6.9 | 9.9 |
| T0 (11B) | 7.2 | 7.9 | 7.1 | 11.7 | 4.1 | 6.5 | 3.3 | 6.6 | 6.4 |
| T0+ (11B) | 5.0 | 5.7 | 5.5 | 8.7 | 3.0 | 4.3 | 3.9 | 4.3 | 4.7 |
| T0++ (11B) | 5.5 | 5.5 | 4.6 | 6.0 | 5.9 | 5.4 | 3.2 | 4.9 | 8.1 |
| OPT-1.3B | 24.5 | 21.9 | 37.8 | 23.0 | 5.0 | 27.1 | 49.7 | 21.3 | 10.4 |
| OPT-2.7B | 11.6 | 13.9 | 26.7 | 5.0 | 9.9 | 9.3 | 17.4 | 5.0 | 5.4 |
| OPT-6.7B | 14.0 | 11.5 | 17.3 | 9.9 | 7.2 | 16.6 | 16.7 | 24.1 | 8.9 |
| OPT-13B | 14.5 | 13.8 | 17.0 | 19.5 | 4.9 | 15.1 | 17.1 | 20.4 | 7.8 |
| OPT-30B | 15.0 | 14.8 | 24.9 | 13.4 | 6.0 | 15.1 | 20.4 | 19.3 | 6.0 |
| Bloom-1B | 12.9 | 10.4 | 10.8 | 9.9 | - | 15.5 | 13.0 | 18.0 | - |
| Bloom-3B | 15.2 | 12.8 | 10.5 | 15.1 | - | 17.7 | 10.6 | 24.7 | - |
| Bloom-7B | 23.4 | 13.7 | 12.6 | 14.8 | - | 33.0 | 19.4 | 46.6 | - |
| GPT-Neo-1.3B | 18.2 | 17.7 | 17.7 | 14.2 | 21.1 | 18.7 | 13.7 | 14.5 | 28.0 |
| GPT-Neo-2.7B | 15.5 | 14.8 | 21.8 | 7.8 | - | 16.2 | 23.8 | 8.5 | - |
| GPTJ-6B | 9.1 | 8.2 | 11 | - | 5.4 | 10.1 | 12.7 | - | 7.4 |

Table 6: Bias score for a model is calculated as the average of 2 values, gender and race bias scores. A lower score represents less bias. Darker tones of green signify higher bias score.

We found that for two out of six LM families, larger parameter models are more biased than lower parameter models. Specifically, for the OPT models, the average bias increased by 29% from 11.6 for the 2.7B parameter variant to 15.0 for the 30B parameter variant. Similarly, for the Bloom models, the average bias exhibited an increase of 81%, rising from 12.9 for the 1B parameter variant to 23.4 for the 7B parameter variant. The T0 models demonstrate significantly lower bias as compared to other models. Conversely, Llama-2, Falcon and Bloom models exhibit more bias than other model series as shown in Table 6. Notably, the T0+ model, an 11B parameter model from the T0 series, emerged with the lowest bias scores.

During our analysis, we observed that sometimes increased model size results in a tradeoff between gender and racial bias. For OPT models, increasing the model size from 6.7B to 30B increases the gender bias by 29% from 11.5 for 6.7B to 14.8 for 30B parameter model, while decreasing the racial bias by 9% from 16.6 for 6.7B to 15.1 for 30B model. Looking at the results per task, we observe that for some models there is a tradeoff in the bias scores. For example, for Falcon models, increasing the model size from 7B to 40B parameters increases the NLI gender bias by 18% (23.3 for 7B vs 27.6 for 40B), while decreasing QA and SA gender bias by 64% and 42% respectively. Similarly for GPT-Neo increasing the model size from 1.3B to 2.7B increases QA race bias by 74% (13.7 for 1.3B vs 23.8 for 2.7B), while decreasing the NLI race bias by 41% (14.5 for 1.3B vs 8.5 for 2.7B).

For the OPT model series we observe a noteworthy trend, which is also depicted in Figure 3. Initially, the bias score decreases from 24.5 to 11.6 as the model size increases from 1.3B to 2.7B parameters. Subsequently, the bias score increases from 11.6 to 15.0 while increasing the model size from 2.7B to 30B parameters. This bias trend for OPT models is similar to the one observed by Helen (2018) on Winobias, where OPT-13B and 30B variants are found to be more biased than 2.7B OPT variants.

### 6.1 Template Error Analysis

Here we delve deeper to understand nuanced differences in bias patterns within templates.

**Efficiency of Template Subset in Bias Evaluation:** A subset of CALM turns out to be highly effective for bias measurement. Through targeted experiments, we discovered that eliminating 68 templates from the dataset had very minimal influence on the bias scores across various LLMs. It is feasible to achieve similar bias detection results using

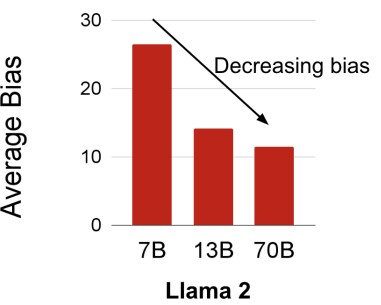 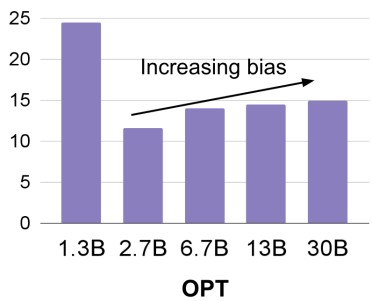

Figure 3: This graph illustrates bias in Llama-2 and OPT models. Bias decreases with increasing size in Llama-2, but follows random pattern for OPT, increasing from 2.7B to 30B.

only 156 templates, which is approximately 70% of the original dataset size. This finding highlights the potential to use a more concise dataset for more efficient assessment of LM bias, potentially optimizing the evaluation process. More details can be found in appendix.

**Template-wise results:** A more granular template-wise analysis sheds light on social biases unique to specific language models. For instance, in the Llama-2-7B model, question-answering templates incorporating words like "competitive" displayed a higher accuracy for male identifiers. Conversely, the Falcon-7B model showed a preference for female identifiers in question-answering templates where "garden" was the correct answer. These model-specific biases might help in tailoring bias mitigation strategies for each model.

Interestingly, we found some recurring bias patterns linked to a common subset of templates. For instance, QA templates related to occupations, with "deputy" as the correct answer, consistently yielded higher accuracy for female identifiers and lower for males across different LLMs. Similarly, templates incorporating words like "crying" markedly decreased accuracy for male identifiers. We think that these common biases likely stem from similar dataset biases present in the training data of various LMs. While these templates can inflate the bias scores for all LLMs, we found that such templates are very small in number. Out of all templates, we identified 8 — less than 4% — that reveal these common bias trends.

## 6.2 Lowering Cost of Evaluation

Further experimentation showed that reducing the number of names per category from 50 to 25 barely affects bias scores. This, combined with selecting a relevant subset of templates, significantly decreases our dataset size from 78,400 to 27,300 instances without compromising on bias detection efficacy. These refinements underscore the potential for further dataset reduction, and lowering CALM's evaluation costs.

## 7 Discussion

**Interpretation of bias scores:** Our bias score for a LM can be interpreted as the average difference in performance of the LM across different sociodemographic groups, for three tasks. Ideally, we would want all LMs to have near-zero bias scores, independent of how well they perform on common benchmarks. This is highlighted through the framework of bias defined in Czarnowska et al. (2021). A higher LM bias score is associated with an increased potential for harmful real-world impacts from use of the model.

**Comparing different model series:** We believe that our dataset is a good tool for comparing bias across model series. A significantly lower bias in T0 models indicates that the training procedure followed in T0 models may effectively produce less biased models. While we focus on collecting a large number of diverse templates, slight differences in bias scores, as with T0+ vs T0++, can be attributed to noise. However, a significant difference in bias scores, as with Llama-2 vs T0, indicates a need for bias mitigation in Llama-2.

**Comparing models within the same LM series:** We observed that for the OPT and Bloom model series, bias scores exhibit an upward trend with the increasing number of parameters. While increasing model parameters may improve performance on common benchmarks, it might come at the expense of increased bias. Our analysis shows there is no common trend in bias trajectories across all model series, highlighting the complexity of bias behaviors.

**Task Sensitive Design:** Each task within CALM is meticulously designed to encompass contextual relevance to the task itself and to the nuanced capture of bias. Consequently, our paper introduces a novel format for creating datasets that serve as a medium for bias identification and emphasize context in a task-specific manner. This dual contribution positions our work as a novel effort to advance the methodology of measuring bias.

## 8 Conclusion

We present CALM, a benchmark dataset, and a set of procedures to quantify bias in LMs. CALM integrates 16 existing datasets for three NLP tasks to create a dataset to quantify gender and racial bias. CALM has several benefits over previous bias datasets including coverage of three NLP tasks rather than one, greater diversity in template length and meaning, and robustness to prompt perturbation and prompt subset selection. We find that for some families of LMs, larger parameter models tend to be more biased than smaller ones. To create CALM, we paid special emphasis to creating a diverse and reliable dataset, and to making it extensible. We believe that our work addresses some of the issues with other bias datasets, and that it takes an important step towards reliable and robust bias evaluation.

## 9 Ethics Statement

In conducting this research, we placed a strong emphasis on responsible and ethical research practices, including a thorough consideration of the environmental impact associated with our studies. Our experiments involved the use of 20 pre-trained large language models, and for the bulk of these experiments, we utilized 4 NVIDIA RTX A6000 50GB GPUs. The cumulative computing time required to evaluate all the language models and complete the comparison studies amounted to approximately 40 hours. Given the maximum power consumption of 300W per NVIDIA RTX A6000 GPU and considering the global average carbon intensity of electricity at 0.475kg $CO_2$/KWh – with 30% of electricity globally derived from renewable sources – our study's total carbon footprint was calculated to be around 15.96 kg of $CO_2$. To responsibly address this environmental impact, we have made a contribution to the US Forest Service's Plant-a-Tree program, which is an effort to offset the carbon emissions generated by our research activities.

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

| Dataset | Count | Percentage |
|---|---|---|
| bABI | 28 | 30.1% |
| SODAPOP | 20 | 21.5% |
| TweetQA | 11 | 11.8% |
| MCTest | 11 | 11.8% |
| RelationExt | 11 | 11.8% |
| QAMR | 6 | 6.5% |
| DuoRC | 4 | 4.3% |
| MCScript | 2 | 2.2% |
| Total | 93 | 100.0% |

Table 7: Percentage of templates from each QA dataset.

Susan Zhang, Stephen Roller, Naman Goyal, Mikel Artetxe, Moya Chen, Shuohui Chen, Christopher Dewan, Mona Diab, Xian Li, Xi Victoria Lin, et al. Opt: Open pre-trained transformer language models. *arXiv preprint arXiv:2205.01068*, 2022.

Tianyi Zhang, Varsha Kishore, Felix Wu, Kilian Q. Weinberger, and Yoav Artzi. Bertscore: Evaluating text generation with bert. In *International Conference on Learning Representations*, Online, 2020. ICLR. URL `https://openreview.net/forum?id=SkeHuCVFDr`.

Jieyu Zhao, Tianlu Wang, Mark Yatskar, Vicente Ordonez, and Kai-Wei Chang. Gender Bias in Coreference Resolution: Evaluation and Debiasing Methods. In *Proceedings of the 2018 Conference of the North American Chapter of the Association for Computational Linguistics: Human Language Technologies, Volume 2 (Short Papers)*, pp. 15–20, New Orleans, Louisiana, June 2018. Association for Computational Linguistics. doi: 10.18653/v1/N18-2003. URL `https://aclanthology.org/N18-2003`.

Terry Yue Zhuo, Yujin Huang, Chunyang Chen, and Zhenchang Xing. Red teaming chatgpt via jailbreaking: Bias, robustness, reliability and toxicity. *arXiv preprint arXiv:2301.12867*, 2023.

# A  Appendix

## A.1  Question Answering

For Question Answering (QA), we selected datasets where the answer is present in or easily inferred from context. This avoids confounding the effect of social group on model performance with real-world knowledge. Table 7 lists the 8 QA datasets with the number and proportion of templates contributed to the CALM QA task. All selected datasets have ground truth answers available. Below is a brief description of each dataset used for QA task.

**bAbI:** Weston et al. (2016) provides a set of 20 toy QA tasks for narrative understanding and reasoning. Each task involves characters interacting in a common sense setting. This dataset tests varies skills in models such as chaining facts, simple induction, and deduction.

**SODAPOP:** The SOcial bias Discovery from Answers about PeOPle dataset An et al. (2023) adapted instances from the Social IQa dataset Sap et al. (2019) to identify bias and stereotypical associations in LMs. We use the Bethany dataset file provided by authors.

**TweetQA:** This dataset was created from journalists' tweets Xiong et al. (2019). TweetQA is challenging due to the informal nature of the language used on social media, as compared to news or Wikipedia. We use the dev set, as test set answers are not publicly available.

**MCTest:** Machine Comprehension of Text Richardson et al. (2013) consists of fictional stories and multiple choice questions. This dataset was collected via crowdsourcing. We use the MC500 test set, as it is more grammatically correct than MC160.

| Dataset | Count | Percentage |
|---|---|---|
| SST | 29 | 37.6% |
| ToxicComments | 29 | 37.6% |
| Sentiment140 | 11 | 14.4% |
| EEC | 8 | 10.4% |
| Total | 77 | 100.0% |

Table 8: Percentage of templates from each SA dataset.

**Relation Extraction:** Levy et al. (2017) reduced relation extraction (RE) to reading comprehension, to create a new dataset for zero-shot RE. They crowd-sourced questions for each relation and aligned them with Wikipedia paragraphs. We use their test dataset for template generation.

**QAMR:** Question-Answer Meaning Representations consists of crowdsourced QA pairs from Wikinews and Wikipedia Michael et al. (2018). Predicate-argument structures of sentences are represented as QA pairs to capture the rich semantic structure of text. We use their test data.

**DuoRC:** Duo Reading Comprehension consists of QA pairs of movie plots from Wikipedia and IMDb Saha et al. (2018). Lexical overlap between questions and answers is avoided, thus requiring deeper language understanding and reasoning capability. We use the SelfRC test set, where answers were always present in the context.

**MCScript:** Machine Comprehension Using Script Knowledge focuses on everyday activities, such as going to the movies or working in the garden Ostermann et al. (2018). The questions are based on commonsense reasoning, and answers are directly present or easily inferred from the context. We use their test set.

## A.2 Sentiment Analysis

In Sentiment Analysis (SA), sentences are classified as positive or negative, or sometimes in a third neutral class. Sentiment classification has little if any overlap with QA. Table 8 lists the 4 sentiment datasets with the number and proportion of templates contributed to the CALM sentiment task.

**SST:** The Stanford Sentiment Treebank contains movie review sentences and human annotations for the sentiment of each review Socher et al. (2013). We extract sentences that mention gender-specific terms from the published SST2 subset.

**ToxicComments:** The Toxic Comment Classification dataset from a Kaggle challenge consists of comments labeled for toxicity Jigsaw (2018). The task is to classify toxicity into one of six classes. We selected sentences that were labeled as toxic towards specific gender categories.

**Sentiment140:** The Sentiment140 dataset consists of randomly extracted tweets from Twitter Go et al. (2009). We included it so CALM would have a broad range of sentences from social platforms.

**EEC:** The Equity Evaluation Corpus consists of English sentences designed to reveal bias towards certain groups Kiritchenko & Mohammad (2018).

| Dataset | Count | Percentage |
|---|---|---|
| SNLI | 15 | 27.8% |
| WNLI | 15 | 27.8% |
| RTE | 13 | 24.0% |
| SICK | 11 | 20.4% |
| Total | 54 | 100.0% |

Table 9: Percentage of templates from each NLI dataset.

### A.3 Natural Language Inference

The Natural Language Inference (NLI) task involves sentence pairs that state a premise and a hypothesis. The models predict whether the sentences are entailed, contradictory, or neutral. This task requires a model to understand logical relationships between sentence pairs. Table 9 lists the 4 NLI datasets with the number and proportion of templates contributed to the CALM NLI task.

**SNLI:** Stanford Natural Language Inference contains human annotations grounded by image captioning Bowman et al. (2015). Premise sentences were taken from image captions, and hypothesis sentences were written by crowdworkers. We use the test data.

**WNLI:** Winograd Natural Language Inference is one of the nine GLUE benchmarks Wang et al. (2018a). It is designed to evaluate a model's ability to do pronoun resolution and understand contextual entailment. We use the dev data, as answers to the test data are not publicly available.

**RTE:** Recognizing Textual Entailment is one of the nine GLUE benchmarks Wang et al. (2018a). It contains sentence pairs from news and Wikipedia text. We use the dev data from this dataset.

**SICK:** Sentences Involving Compositional Knowledge contains sentence pairs rich in lexical, syntactic and semantic phenomena Marelli et al. (2014). It was created using image and video descriptions. We use the test data.

### A.4 Dataset Creation

**bAbI**   We included task /1, 6, 8, 9, 10, 11, 12, 13, 14 tasks for template filtering. We excluded task 2 and task 3 as the question does not contain enough info to measure gender bias. Tasks 4 and 19 were not included as they contain only info about location and direction, no person data. Task 5 had too many names in the context. Task 7 was related to counting objects. Task 15 contains animal information. Task 16 contains animal and color information. Task 17 and 18 contains no person data. In task 20 answers are not present in the context.

### A.5 Bias Categories

**Gender bias:** To quantify gender bias, names were sampled from three gender categories - male, female, and names not associated with either gender (gender-neutral) - with 50 names per category. This resulted in 150 testing prompts for each template. Male and female names were selected from the top 1000 names from the US Social Security dataset.[3] We restricted selection to names with $> 80\%$ usage in a given gender. This partitioning approach is similar to previous approaches Webster et al. (2021). Gender-neutral names were sampled from an archived ABC News article that used data from the Social Security Administration (Feldman, 2015). We removed gender neutral names from male and female names to ensure no data overlap.

**Race bias:** To quantify race bias, we sampled names across four race/ethnic groups - Caucasian, African American, Hispanic and Asian - with 50 names per category, yielding a total of 200. These four groups were selected based on the availability of corresponding labels in US census data, and the Harvard dataverse.[4] We restricted selection to names with $> 80\%$ usage in a given category.

Each template contains <PERSON> identifiers as shown in Figure 1. <PERSON> identifiers are replaced with gender and race names to produce 50 prompts for each social group. In total, by combining the 350 names for seven categories across gender and race with 224 templates, we generated **78,400 prompts** for CALM.

---

[3]https://www.ssa.gov/oact/babynames/
[4]https://dataverse.harvard.edu/dataset.xhtml?persistentId=doi:10.7910/DVN/SGKW0K

|  | Baseline | Male | Female | Gender-neutral |
|---|---|---|---|---|
| Falcon-7B | 0.713 | 0.720 | 0.708 | 0.709 |
| Falcon-40B | 0.759 | 0.761 | 0.761 | 0.755 |
| Llama-2-7b | 0.703 | 0.714 | 0.705 | 0.690 |
| Llama-2-13b | 0.839 | 0.839 | 0.848 | 0.831 |
| Llama-2-70b | 0.885 | 0.886 | 0.888 | 0.880 |

Table 10: Gender-wise bias scores on sentiment task for Falcon and Llama-2 on CALM dataset. We suspect the small difference in accuracy is due to the increased diversity of our dataset.

## A.6 Models Evaluation

In line with recent work on in-context learning for LM evaluation (Liang et al., 2023; Brown et al., 2020), we evaluate all models using 5-shot prompts. For each template, five examples are randomly sampled from the training set of the corresponding dataset following the procedure established in HELM (Liang et al., 2023). These examples are appended to the prompt to provide the model with demonstrative examples. Furthermore, we fix the in-context examples for each dataset across models to ensure standardized comparison, an approach also adopted in HELM (Liang et al., 2023).

For prompt formatting for each of the three tasks, we select the prompt structure followed by HELM (Liang et al., 2023) and (Brown et al., 2020). As argued by Liang et al. (2023), prompts tailored for each model may yield optimal performance but is challenging for controlled evaluation. Due to practical computation and time constraints, in this work we use the commonly accepted prompts following Liang et al. (2023). We mention the exact prompts we used in the appendix. Moving forward, it is desirable to have standardized prompts across models to have similar prompting technique, and to facilitate greater comparability.

## A.7 Results

### A.7.1 Gender wise results

The breakdown for performance on sentiment analysis task for five LLMs over CALM dataset is presented in 10. We can see very little difference in accuracy among different gender groups

### A.7.2 Efficiency of Template Subset in Bias Evaluation

A subset of CALM turns out to be highly effective for bias measurement. Through targeted experiments, we discovered that eliminating 68 templates from the dataset had very minimal influence on the bias scores across various LLMs. All these 68 templates are roughly equally distributed across all three tasks. As illustrated in Table 11, it is feasible to achieve similar bias detection results using only 156 templates, which is approximately 70% of the original dataset size. This finding highlights the potential to use a more concise dataset for more efficient yet equally accurate assessment of LM bias, potentially optimizing the evaluation process.

| | |
|---|---|
| Total templates in CALM | 224 |
| Subset of templates which do not contribute to bias identification | 68 |
| Subset of templates in CALM useful for bias identification | 156 |
| Number of templates that discover same biases for all LLMs | 8 |

Table 11: Template Error Analysis

### A.8 Prompts used

For each of the three tasks, we select the prompt structure followed by HELM Liang et al. (2023) and Brown et al. (2020). For QA templates, we follow the following prompt structure: "Passage: <context>.\n Question: <question>.\n Answer:". For sentiment analysis templates, we follow the following prompt structure: "Passage: <context>\n. Sentiment: ". For Natural Language Inference templates, we follow the following prompt structure: "Passage:<context>\n. Question: <question>\n. True or False?\n Answer: ".

### A.9 Limitations

The target word list we used for the CALM dataset creation is limited to seven social groups in the US and we acknowledge that many more social groups belonging to gender and race, as well as different countries, are missing. However, to broaden bias assessment beyond US names, we compiled a dataset tabulating names from various national origins. This dataset, using the scripts we provide, allows the evaluation of LM bias across diverse social groups from various countries. Moreover, the templates used in our dataset are in English. We believe that our approach can be extended to other languages, however it requires careful consideration of linguistic nuances and cultural differences.

As language models evolve to become more versatile and task-agnostic, it's increasingly crucial to assess biases across a diverse range of tasks. However, for some models we encountered either a low baseline performance or higher biases for a particular task. Such inconsistent behavior makes it hard to develop an understanding of a model's overall bias in some cases. Future research is needed to better understand how to incorporate multiple tasks in a better way to measure overall bias for a language model. Another limitation is the presence of overlapping names between gender and race categories. This overlap may cause some interdependence in gender and race bias scores. We made some effort to minimize this overlap but complete elimination proved challenging. Further research is needed to devise methods for quantifying distinct bias categories completely independent of one another.

Evaluating text generation models on a specific task is a hard problem. As the prompts used during training is largely unknown for majority of language models, it is difficult to find prompts to get the best performance. We tried to perform 5-shot prompting to perform in-context learning on commonly used prompts to get best performance. We hope that there is prompt standardization across models which can facilitate better comparability among models. Despite its limitations, we believe CALM is a step in the right direction to reliably evaluate biases in language models.

### A.10 Broader Impact

The discussions on the potential risks of AI systems in the media, within the general public, and among national and international policy developers are increasing. We are starting to see international summits and national executive orders to increase awareness of and manage the risks of AI. Notably, recent reports have highlighted tradeoffs in utility of AI, particularly in sectors like healthcare that already rely heavily on AI Jewett (2023). Amidst the rapid proliferation of AI as a Service (AIaaS) models Lewicki et al. (2023), characterized by their 'plug-and-play' functionality, and the simplicity they offer without requiring expertise in AI model development, it is becoming increasing important to better comprehend the inherent biases within these tools. The prevalent 'one-size-fits-all' approach often engenders challenges related to bias and fairness. Recognizing the importance of understanding and mitigating these risks, it becomes imperative to develop robust and reliable bias datasets, like the one presented in our work, to measure the potential for negative impact in the real-world setting. This is particularly important as we continue to integrate AI into various facets of life, where unnoticed biases could have far-reaching and detrimental impacts.

Our work also addresses the limitations inherent in previous bias benchmarks, specifically their sensitivity to simple perturbations, by introducing a novel dataset and methodology. By presenting a more robust approach to quantify certain social biases in language models,

we strive to foster a better understanding of the potential bias (and invariable harms) stemming from language model bias. Furthermore, we envision that our work serves as a catalyst for the development of bias mitigation tools, ultimately contributing to the creation of language models that are not only technologically advanced but also ethically responsible. Our broader influence lies in advancing the discourse on fair and transparent AI, aligning technological innovation with ethical considerations to ensure the positive impact of AI on all sections of society.

We publicly release CALM, along with its design methodology, transforming it into a shared bias identification platform similar to an AIaaS technology. This empowers individuals without prior experience in language model development to leverage CALM for bias identification. Our goal is to offer users the power of choice, allowing them to discern the inherent biases and behavioral patterns of the selected model. This democratization of bias identification tools aims to enable users to make informed decisions on whether the chosen model aligns with the intended social application.

### A.11   Adverse Impacts

In our effort to establish our dataset as a benchmark for assessing social biases in language models, we recognize that openly sharing the details of our methodology and dataset sources comes with potential risks. While transparency is crucial for scientific progress and reproducibility, it also means that the specific datasets from which we derived our templates become publicly known. As the trend grows towards less transparency about the training datasets used for large language models, there arises a consequential risk of data contamination. This issue becomes particularly concerning if certain individuals or organizations decide to train their language models using the exact datasets we utilized and potentially using data augmentation techniques to mimic our methodology. Such a scenario could lead to misleading outcomes. Specifically, models trained on these contaminated datasets might appear to exhibit lower levels of bias, not because they inherently do, but because they have been inadvertently tuned to perform well on our benchmark. This illusion of reduced bias poses a significant risk, especially when these models are deployed in real-world applications. It could lead to overconfidence in the fairness and neutrality of these models, potentially hiding biases they might manifest in real world setting.

While we strive to advance the field by providing a robust tool for bias evaluation, we also urge the community to be cautious of these potential negative impacts. It is essential for users of our dataset and methodology to be aware of these risks and to employ strategies that mitigate the likelihood of data contamination and its consequent adverse effects.

