# OpenReview forum: "CALM : A Multi-task Benchmark for Comprehensive Assessment of Language Model Bias"
_colmweb.org/COLM/2024/Conference — COLM_

### Official Review · Reviewer_SNfb · 2024-04-25

**Rating:** 6
**Confidence:** 3
**Ethics Flag:** 1

**Summary:**

This paper introduces a new benchmark to assess the presence of gender and race bias in LMs.
The main novelty lies in the high number and diversity of templates for these two bias categories. This is achieved by selecting templates from three existing datasets covering the tasks of QA, sentiment analysis, and NLI, respectively.

The benchmark is used to quantify bias in 20 open-source LLMs.

**Questions To Authors:**

OTHER COMMENTS:

"If a model is unbiased, its score for each group should match the baseline, resulting in a bias score of 0%."
=> I'm not convinced this is the case. This is because each group accuracy is computed on a different, smaller set of instances than the baseline score which is computed on all instances. It would be more reasonable to expect that, in a less biased model, the group-specific scores will approach a baseline score averaged over several similar-size, randomly sampled subsets of the entire dataset.
In any case, expecting a difference of 0 seems unrealistic.
Some testing of statistical significance is advised especially when differences among models' bias scores are small.

The comparison to the BBQ dataset (the largest of existing bias benchmarks cited in the paper) could be more specific.

Examples of observations on the results that seem a bit shaky:
-  for OPT series, larger model means more bias: This is only true if you ignore the smallest OPT which has by far the largest bias in this series. The other differences may not even be statistically significant.
- tradeoffs between gender and racial bias: Sometimes gender bias is higher in a larger model version, sometimes race is. There's a big chance this is a random effect, possibly based on idiosyncrasies of the datasets on which these models were trained. Why call it a "tradeoff"?
- tradeoffs between bias in different tasks: Idem.

The template error analysis focusing on a few cherry-picked words (e.g. "competitive" associated with male bias) doesn't seem very insightful.

Typos:
- Prior work on bias measurement are -> is
- Construction of CALM -> The construction of CALM

Clarity:
- Many appendices are included, but all references from the main body point to a generic "Appendix"
- some appendices (...) provide hardly more information than what's already in the main body
- explain what HELM does

**Reasons To Accept:**

The new benchmark can be useful to complement the existing ones, and contribute to a more reliable evaluation of gender and race bias in LMs thanks to its wide range of templates.

**Reasons To Reject:**

Reproducibility can be improved, especially w.r.t. the template selection process, the template modifications to assess robustness, and the section on the template subset)

While the benchmark can be clearly useful, the analysis conducted on top of it leaves space for improvement. Specifically:
- Statistical significance of bias score differences is never mentioned
- Many patterns are drawn from many bias score differences that could just be due to noise. For instance, many observations are made on the effect of model size, but as it stands these differences could be due to different contents of the datasets on which different-sized models were trained. Similarly, it is suggested that T0's lowest reported bias score could be due to its training procedure, but here too dataset content seems to be a more natural explanation

[Not per se a reason to reject, but reflects the general lack of substance in the paper]:
- much space is dedicated to repeating how this benchmark is better than others (e.g. Figure 2 and much of section Discussion), instead of letting the readers judge this by themselves based on the content of the paper. Some points of strengths are overstated (e.g. yes there are more templates in this work, but other works had a larger variety of bias categories which is also important). In my opinion it's more important to convincingly show complementarity wrt previous work, rather than overstating superiority.

---

> ### Author Rebuttal · Authors · 2024-05-31
>
> We thank the reviewer for their detailed review. We appreciate the reviewer’s acknowledging the reliability of our dataset and the diversity of templates used. Below we address the concerns raised :
>
> 1. We acknowledge the importance of statistical significance in our analysis. To address this, we ran subsets of CALM across multiple runs to check the variance in bias scores, as described in Section 4.2. Our results showed minimal variation, indicating low noise. We agree that small changes in our bias scores can be attributed to noise, as also mentioned in paragraph 2 of section 7. Therefore, we only draw conclusions when bias score differences between models exceed 10% (relatively), based on the maximum deviation observed in Table 3. Thus for OPT models, the shown trajectory is relevant and should not be attributed to noise as we belive a 29% jump (11.6 for the 2.7B vs 15.0 for the 30B) in bias scores is significant, specially when considering these models for real-world use cases. Similarly, we discuss patterns for tradeoff in gender vs race, and different tasks only when the difference in bias scores was more than 10%.
> 2. Our intention was not to overstate superiority but to demonstrate, with extensive experimentation, the robustness and reliability of our dataset compared to prior works, a point echoed by Reviewer B3CC and Zsmu. Our intention with this work is to highlight a new methodology for robust and reliable measurement of bias in LMs. Our methodology for finding diverse templates is complementary to other works as also acknowledged by Reviewer B3CC, and can be easily expanded to more bias categories and used by future works to develop reliable metrics on new bias categories. We apologize if our language suggests otherwise and will ensure clarity in our language during the camera-ready revision, along with a better explanation for BBQ and HELM
> 3. We also notice that you have flagged an issue regarding reproducibility. We would like to highlight that we will be releasing our complete codebase which will include detailed steps for template extraction, modifications, and scripts to run all experiments. This will ensure that our methodology can be replicated by other researchers.
>
> We have tried to address all the concerns raised by the reviewer and we believe our work aligns perfectly with the conference theme. We hope the reviewer will reconsider the evaluation of our work, providing strength and confidence to the rebuttal process.

---

> > ### Comment · Reviewer_SNfb · 2024-06-04
> > **Thanks for your answers. I still find some of the conclusions stretched.**
> >
> > Thanks for your answers. I still find some of the conclusions stretched.
> > For instance, I explicitly mentioned this one:
> > “for OPT series, larger model means more bias: This is only true if you ignore the smallest OPT which has by far the largest bias in this series. The other differences may not even be statistically significant.”
> > but the authors did not comment on it.
> > To be clear the smallest OPT model has a bias score of 24.5 versus all the other models have scores between  11.6 and 15.0. This strongly undermines the conclusion that larger parameters lead to larger bias.
> >
> > The author response also does not address several of my other comments, for instance:
> > - "If a model is unbiased, its score for each group should match the baseline, resulting in a bias score of 0%." => I'm not convinced this is the case.
> > -  it is suggested that T0's lowest reported bias score could be due to its training procedure, but here too dataset content seems to be a more natural explanation

---

> > > ### Author Response · Authors · 2024-06-04
> > > **Thank you for the comment, we provide further explanation**
> > >
> > > We thank the reviewer for their comment. We provide further explanation below.
> > >
> > > Regarding trend in OPT series, we agree that OPT-1.3B has a higher bias and doesn’t follow the upward bias trend. In the paper, we mentioned this in paragraph 4 of section 3 and we claim for OPT models to have higher bias trend with growing model size from 2.7B to 30B parameters. The point we are trying to make is that bias scores can increase with increasing model sizes. We are not trying to state that all OPT series models follow the upward bias trend. We apologise for any confusion and will make the text clearer to avoid any confusion in the camera-ready version.
> > >
> > > Regarding the statistical significance of the upward bias trend from 2.7B to 30B version, we believe that a 29% jump in bias scores (11.6 for the 2.7B vs 15.0 for the 30B) is statistically significant as we argued in the rebuttal. To make it clear, we consider all differences higher than 10% (relatively) to be statistically significant based on our extensive experimentation in Table 3. During our analysis, we did not draw any conclusion whenever the bias differences were less than 10% (relatively). The results we observed for upward bias trend in OPT models from 2.7B to higher parameter models is consistent with observations by Helen et al [1].
> > >
> > > Regarding “unbiased model having 0% bias score”, please note that while we are measuring a group accuracy on a smaller set of instances, the only difference with other subsets is change of name in our setting. In an ideal scenario, we would expect models (which are completely unbiased) to generate responses which are not affected by change of names. This reasoning of the expectation of 0% bias scores in unbiased models is consistent with other related works on bias datasets [2,3,4]. We understand that this might seem unrealistic to some readers and we will tone down this sentence to state that we expect the model to have similar accuracy on all subsets.
> > >
> > > During our experimentation, we tried our best to ensure that all the models evaluated have not seen any of the datasets before. Specifically, in the case of T0, only DuoRC (which is <1% of our dataset) was part of training of the T0 series of models. This ensures that there was no prior data contamination which could have significantly impacted our analysis on T0. While, we accept that there might be other factors apart from training procedure which can have resulted in lower bias scores in T0 models, but the bias scores for T0 models are significantly lower than any other tested model series and we believe their training procedure definitely contributed to this lower bias score.
> > >
> > > We hope our responses have addressed the reviewer’s concern and we would be very happy to provide further information. We hope the reviewer will consider our work to be impactful and useful for the community, and will reconsider evaluation of their score.
> > >
> > > References (full references in main paper):
> > > [1] Helen. Very large language models and how to evaluate them, 2018.
> > > [2] Haozhe et al. "SODAPOP". EACL 2023
> > > [3] Tao et al. "UNQOVER". EMNLP 2020
> > > [4] Smith et al. “I’m sorry to hear that". EMNLP 2022

---

### Official Review · Reviewer_B3CC · 2024-05-10

**Rating:** 8
**Confidence:** 4
**Ethics Flag:** 1

**Summary:**

This paper proposes a bias evaluation method (and dataset) called CALM, which builds off the idea of existing template-based evaluations, but greatly increases the number and variety of templates, resulting in a broader evaluation. Furthermore, the authors find that their evaluation is markedly less sensitive to certain perturbations that have previously been shown to drastically affect the results of previous methods, indicating that those other methods are too unstable.

In terms of experiments, the authors assemble their set of templates from 16 existing QA, sentiment analysis, and NLI datasets, identifying person mentions and pronouns where they appear in various instances in those datasets. They then consider various fill-ins for those names and pronouns along racial and gender axes; for race, they use names from US census data and the Harvard dataverse for Caucasian, African American, Hispanic, and Asian people, and for gender, they use male and female names sampled from the US Social Security dataset, and also gender-neutral names sampled from an article listing many such names.

The authors evaluate 20 different existing language models for bias using CALM, present their findings, and close their experimental results with a discussion of how bias relates to sizes of different model series.

**Questions To Authors:**

(1) Just to check, the average bias score for a single model is computed as an average of the race and gender per-template bias scores, right? In other words, the average bias score for a model is computed by first concatenating the lists on the two following lines, and then averaging that concatenated list, right?

$$\left[ \left( \max_{r \in \textrm{4 race categories}}bs(r, t) - \min_{r^* \in \textrm{4 race categories}}bs(r^*, t) \right) \mid t \in \textrm{templates}\right]$$
$$\left[ \left( \max_{g \in \textrm{3 gender categories}}bs(g, t) - \min_{g^* \in \textrm{3 gender categories}}bs(g^*, t) \right) \mid t \in \textrm{templates}\right]$$

If so, great-- that was my reading of the paragraph directly below equation 1. But if it's calculated some other way, please let me know! And regardless, it would be helpful if you could reword the paragraph under equation 1 (or add a formal mathematical expression for the model-level bias score) to clarify. Thanks!

(2) I'm not sure exactly what implication you're driving towards with the point communicated in table 5 (and in the first paragraph of 4.4) about consistent averaged accuracy scores. Is your reading of the results essentially a warning not to just compare aggregated per-category accuracies, since those can obscure important inter-category differences? If so, it would be nice to clarify that.

(3) What were the "targeted experiments" described in the first part of section 6.1? How did you end up deciding to remove those 68 templates in particular?

(4) Which figure is being referenced in the last paragraph of section A.5 in the appendix? Currently the first sentence there ends with "Figure ??."

**Reasons To Accept:**

The way the authors operationalize bias is well-motivated and well connected to previous work.

The authors did a great job of situating their work within the problem space and articulating the key issue with previous work that their work addresses (namely, a lack of robustness in bias evaluation methods).

The experiments involved a wide variety of models, which is great.

Section 4.1 in particular is a really nice experiment that provides a strong argument for CALM's robustness. Borrowing the methods for this analysis from Selvam et al. 2023 that found other methods to be highly sensitive is a nice touch.

On the writing side, the paper's also (generally) very clear; I asked questions about the few parts I found unclear below.

**Reasons To Reject:**

Based on the source datasets used for constructing the templates, it's unclear to me how many of the produced templates are relatively anodyne/wouldn't be expected to correlate with a real-world bias anyway. Essentially, this concern is echoed at the end of section 6.1: "Out of all templates, we identified 8— less than 4%— that reveal these common bias trends" (although in section 6.1, that's framed as a strength). But I'm not too concerned about this, seeing as CALM *does* seem to surface biases in a variety of models. (That said, maybe weaken the description of "criteria directed at socioeconomic distinctions" at the top of page 4 to something like "names and pronouns, since those can carry information about socioeconomic distinctions"-- the current wording makes it sound more like every mention of a name or pronoun that's flagged by your method carries important socioeconomic distinctions, which in a sentence such as "Mary went for a walk," is maybe a bit of an exaggeration.)

The paper's argument for how CALM relates to real-world bias is, as currently presented, pretty weak; in paragraph 2 of section 1, we have "We believe that models having lower CALM bias scores are likely to exhibit reduced biases in practical scenarios," and in section 7, "A higher LM bias score is associated with an increased potential for harmful real-world impacts from use of the model." The first of those sentences is very hedgy, the second of those sentences feels like it should either be citing something or referencing experimental results in order to make that claim, and the paper doesn't actually do any experiment with real-world usage scenarios. I still think the paper overall is valuable for providing a bias evaluation similar to previously proposed bias evaluations that lacks some of their issues, but the current connection to real-world behavior of the models feels too thin. I would recommend either adding some more justification, or removing the discussion of real-world bias as it relates to CALM altogether.

---

> ### Author Rebuttal · Authors · 2024-05-31
>
> We thank the reviewer for their detailed and thorough review. We are very encouraged by your acknowledgment of our comparison and extensive experimentation to address reliability and robustness issues in existing benchmarks. We appreciate the feedback on rewriting some parts of the paper and will make those edits in camera-ready version. Here are the responses to the issues highlighted:
>
> 1. Regarding the templates used in our dataset, we conducted a manual filtering process to ensure they relate to real-world biases. This process helped us maintain high quality and relevance in our bias measurements. When we mentioned, “Out of all templates, we identified 8— less than 4%— that reveal these common bias trends”, we meant that these templates consistently found similar bias patterns across all tested models. While useful for identifying bias, they may not be ideal for comparative analysis as they yield similar bias scores across models.
> 2. We designed this work as an algorithmic way to identify bias in language models. Although we performed some qualitative analysis, a comprehensive examination of how our bias scores relate to real-world scenarios was beyond this study's scope. We intend to do such an analysis in our follow-up work and agree that it will add a lot of value to the field.
> 3. Yes, the average bias score is computed as an average of the race and gender per-template bias scores. We appreciate your feedback on the clarity of this calculation and will revise the paragraph for the camera-ready version.
> 4. In Table 5, we highlight that while aggregate scores across all templates for each category are similar, the per-template differences are substantial. This reflects the high diversity of our dataset, with each template capturing bias in different contexts. We will clarify this point further in the revised paper.
> 5. Our targeted experiments identified templates that are not useful in measuring biases. These are the templates that were either too easy (close to 100% accuracy) or very difficult for all models (close to 0% accuracy). Removing these templates, along with reducing the number of names per category from 50 to 25, reduced our dataset size by 65% without affecting bias scores, thus lowering evaluation costs.
>
> We appreciate the highly constructive feedback and will incorporate these insightful suggestions to improve the clarity and impact of our paper. We genuinely hope the reviewer will reconsider their evaluation based on our rebuttal.

---

> > ### Comment · Reviewer_B3CC · 2024-06-01
> > **Thanks for your response!**
> >
> > Thanks for responding to the points I raised, your answers help to clarify some points I was wondering about. Based on your responses, I'm inclined to leave my score as-is; I still believe the demonstration that the proposed evaluation method is more stable merits acceptance of the paper.

---

> > > ### Author Response · Authors · 2024-06-04
> > > **Thank you for your acknowledgment**
> > >
> > > We thank the reviewer for their score and finding our method to be impactful. We are glad that we could clear some points in our rebuttal.

---

### Official Review · Reviewer_Zsmu · 2024-05-11

**Rating:** 6
**Confidence:** 3
**Ethics Flag:** 1

**Summary:**

The paper introduces the Comprehensive Assessment of Language Models (CALM), a benchmark designed to measure gender and racial biases across 3 tasks -- Question Answering, Sentiment Analysis, and NLI. The benchmark creation process involves selecting popular datasets for each task, generating templates by replacing names and pronouns with a <PERSON> tag, and subsequently replacing the <PERSON> tag with names sampled from different gender and racial groups, resulting in 78,400 examples.

To ensure the robustness of the benchmark, the authors also created four alternative versions of CALM by perturbing the original templates through synonym substitution, addition of clauses, and inclusion of adjectives.
The authors also perform a battery of experiments on 6 LLM families (Llama2, Bloom, Falcon, OPT, T0, GPT-Neo) to validate the robustness of the dataset.

**Reasons To Accept:**

This benchmark covers a variety of tasks and could be useful for measuring the gender and racial bias of LLMs.

**Reasons To Reject:**

While the main contribution lies in introducing this benchmark, the paper lacks a thorough qualitative analysis of the resulting dataset examples. I’m not convinced that simply replacing names in datasets such as SNLI or SST would be sufficient to obtain a dataset to benchmark social biases of strong LLMs like ChatGPT reliably, and it would be good to show experiments on this.

It is unclear whether the prompts mentioned in Appendix A.8 are fit for accessing the model’s performance on some of the chosen datasets. Example:
- Is the template ”Passage:<context>\n. Question: <question>\n. True or False?\n Answer: ”.  fit for evaluating SNLI performance?).
- How was the QA template (”Passage: <context>.\n Question: <question>.\n Answer:”) applied to Multiple Choice QA datasets such as MCScript? This needs clarification.

---

> ### Author Rebuttal · Authors · 2024-05-31
>
> We thank the reviewer for their insightful review and are really encouraged that they recognize the usefulness and extensive experimentation to show the robustness of our proposed dataset and methodology. Below, we provide additional context on the issues mentioned.
>
> 1. In this work we benchmarked powerful LLMs such as Llama-2-70B and Falcon-40B (which were among the top-performing LLMs at the time of this work). As shown in Section 6, these powerful LLMs exhibited social biases for prompts created by name changes, confirming the presence of biases even in advanced models. We understand and support the need for qualitative methods to capture biases but this unfortunately does not fall into the scope of the current work and we intend to pay close attention to the same in our ongoing future work. At the same time, our method of capturing bias is used from an already existing and established literature of template based quantitative analysis, a point echoed by Reviewer B3CC. Our paper enhances the bias measurements by approaching template creation from an unbiased lens.
> 2. We want to highlight we did not directly use datasets like SNLI or SST without modification. Instead, we conducted multiple rounds of filtering to select a small and diverse set of sentences. Our resulting dataset is significantly different than datasets such as SNLI, SST.
> 3. The QA, NLI and SA prompts mentioned in Appendix are adapted from popular works of HELM [1] and Brown et al. [2], which have demonstrated that such prompts generalize effectively across a variety of datasets. While we acknowledge that these prompts may not be perfectly tailored for every dataset and yield sub-optimal performance, customizing prompts for each specific template poses challenges for controlled evaluation, as also discussed in HELM [1]. Please also note that we used 5-shot in-context learning as suggested by HELM [1] to improve model performance.
>
> We have tried to address all the concerns raised by the reviewer and we believe our work is impactful and aligns perfectly with the conference theme. We hope the reviewer will reconsider the evaluation and impact of our work, providing strength and confidence to the rebuttal process.
>
> References (full references in main paper):
> [1] Liang, Percy, et al. "Holistic Evaluation of Language Models." TMLR 2023.
> [2] Brown, Tom, et al. "Language models are few-shot learners." NeurIPS 2020
> [1] Haozhe et al. "SODAPOP". EACL 2023
> [2] Tao et al. "UNQOVER". EMNLP 2020

---

> ### Author Response · Authors · 2024-06-04
> **Request for reconsideration of scores**
>
> Dear Reviewer, given that the reviewer-author discussion phase is coming to a close very soon, we'd greatly appreciate your input on whether there are any pending concerns that we can address which can help in reevaluation of your score.

---

> ### Comment · Reviewer_Zsmu · 2024-06-05
> **Thank you for your response**
>
> Thank you for answering my questions.
> Even if a thorough qualitative analysis is out of scope of the paper, it’d still be good to include examples of more examples of resulting data for each dataset. Although the authors mention that they conducted multiple rounds of filtering to select a diverse set of samples, I’m not convinced this is a good dataset by looking at the examples in Figure 1 and Table 1.
> I’ve the same concern with Reviewer SNfb25 that bias score differences could just be due to noise.
>
> One way to test it is to use a subset of male data as female and vice versa (resulting in computing scores for male vs male, and female vs female) and check if average bias scores for these are close to 0 for all models.

---

> > ### Author Response · Authors · 2024-06-06
> > **New Experiment results**
> >
> > We thank the reviewer for their comment. We ran the experiment suggested by the reviewer. We experimented with 2 models - Llama-2-7B and Falcon-7B. For each model, we performed 4 runs. In 2 runs, we compared performance on a randomly selected subset (20 and 25 male names) with the entire male names set (comparing male vs male scores). For the other 2 runs, we compared performance on a randomly selected subset (20 and 25 female names) with the entire female names set (comparing female vs female scores). For the Llama-2-7B model, we observed an average bias score of 0.3 and for the Falcon-7B model, we observed an average bias score of 0.25. This shows that there is minimal noise in our dataset. Please note that for our analysis, we consider all differences higher than 10% (relatively) to minimise the impact of noise in our metric. This number came from our extensive experimentation in Table 3. We thank the reviewer for suggesting this experiment and this definitely adds value to our paper. We will include this new experiment in camera-ready version.
> >
> > We will add more examples for each dataset in the appendix for a better understanding and will also make all templates accessible along with dataset information in our code release. We hope our responses have addressed the reviewer’s concern and we would be very happy to provide further information. We hope the reviewer will consider our work to be impactful and useful for the community, and will reconsider evaluation of their score.

---

> > > ### Comment · Reviewer_Zsmu · 2024-06-07
> > > **Thank you for your response.**
> > >
> > > Thank you for addressing my concerns. Based on this, I've updated my score.

---

> > > > ### Author Response · Authors · 2024-06-07
> > > >
> > > > Thank you for your response. We are highly encouraged by this fruitful discussion and really appreciate your insightful comments that improves the quality of our work.

---

### Official Review · Reviewer_gvyp · 2024-05-11

**Rating:** 5
**Confidence:** 3
**Ethics Flag:** 1

**Summary:**

The authors construct a template-based bias evaluation dataset based on existing NLI, QA and sentiment task datasets. The dataset focuses on gender and racial bias.

**Questions To Authors:**

- It looks like the evaluation is on a mix of instruction-tuned and pretrained-only models. Should they be treated differently?

**Reasons To Accept:**

The authors repurpose existing task datasets in a unique way to capture the variability of LLM responses across demographic groups.

**Reasons To Reject:**

- If I understand correctly, the proposed bias score primarily measures variability of LLM responses across gender/race-associated names, without capturing any given direction of bias. This is a meaningful distinction from existing bias benchmarks, which often seek to capture positive/negative biases associated with particular demographic groups. While ideally models should respond uniformly across all names, the construction of the current dataset means that higher variability across names, even if not systematically biased, directly translates to a higher bias score.
- The authors make several comparisons to existing bias benchmark in the paper, but based on the above distinction, I do not think that comparisons on e.g. the number of templates is meaningful.

---

> ### Author Rebuttal · Authors · 2024-05-31
>
> We thank the reviewer for their review. We acknowledge the reviewer’s appreciation of our methodology in creating the dataset. Below we provide our response to the issues mentioned:
> 1. Our method was strongly motivated and inspired by the established literature that measures accuracy across names associated with different social groups for bias quantification [1,2,3], also noted by Reviewer B3CC. We would like to emphasize that our major contribution is our methodology to create diverse templates that produce robust measurements. After template formulation, for prompt creation using templates, our approach followed established standards used in prior works. Thus, we believe that our method is not a deviation but rather an enhancement of existing benchmarks, making direct comparisons valid and demonstrating the robustness of our method, ​​a point echoed by Reviewer SNfb, B3CC and Zsmu.
> 2. Gender and race are considered directions of bias in prior works. Our focus in this work is not to capture many bias directions but to provide a new methodology for reliable bias measurements across these 2 bias directions. Our approach for template creation can be easily integrated with other bias datasets to measure various bias directions and we intend to do so in our future works. For example in HolisticBias [4], the 36 simple templates used by the authors can be replaced by templates collected using our methodology, enhancing the reliability and robustness of bias measures proposed in their work. We will make this clearer during the camera-ready revision.
> 3. Regarding the evaluation, all 20 tested LLMs are pretrained-only models. To the best of our knowledge, we did not use any instruction-tuned models, ensuring a fair comparison across LLMs. This was also done in the motivation to audit publicly available systems that can be used by any and all. These systems play a great part in society by transforming LMs to sociotechnical systems.
> We appreciate the constructive feedback and will incorporate these suggestions to improve the clarity and impact of our paper. We genuinely hope the reviewer will reconsider their evaluation based on our rebuttal, providing strength and confidence to the rebuttal process.
>
> References (full references in main paper):
> [1] Haozhe et al. "SODAPOP". EACL 2023
> [2] Tao et al. "UNQOVER". EMNLP 2020
> [3] Prabhakaran et al. "Perturbation Sensitivity Analysis". EMNLP 2019
> [4] Smith et al. “I’m sorry to hear that". EMNLP 2022

---

> > ### Comment · Reviewer_gvyp · 2024-06-06
> >
> > For a quick clarification, I believe T0-3B (Table 2) is instruction tuned?

---

> > > ### Author Response · Authors · 2024-06-06
> > >
> > > We thank the reviewer for their comment. Our understanding is that the usage of term instruction tuning in the current LLM training paradigm is very different from the training procedure used in T0 models. T0 models were trained on a large set of datasets such as WikiQA, IMDB [1]. These datasets were from a variety of different tasks such as QA, summarization etc. To combine different datasets in a single format, the authors added some basic instructions to the prompts such as “Replace the _ in the above sentence with the correct option” [2]. The model was trained only once on this large corpus of datasets [1]. This usage is different from the current LLM training paradigm, where in most cases the training happens in two stages. In the first stage, there is a base model trained on a large corpus of unlabeled data, followed by an instruction-tuning step where models are trained on a set of instructions to give a specified output. In T0 model training, there is no separation between base model and instruction-tuning step. Thus, we mentioned earlier that we did not use any instruction-tuned models.  We thank the reviewer for pointing this out and totally understand that this might be confusing for some readers. We will explicitly mention that the training procedure for T0 models is different from other tested LLMs in camera-ready version. Additionally, to have a fair evaluation between different models, we use in-context learning using 5-shot examples following the approach adopted in HELM [3].
> > >
> > > We hope our responses have addressed the reviewer’s concern and we would be very happy to provide further information. We hope the reviewer will consider our work to be impactful and useful for the community, and will reconsider evaluation of their score.
> > >
> > >
> > >
> > > References (full references in main paper) :
> > >
> > > [1] https://huggingface.co/bigscience/T0
> > >
> > > [2] Sanh, Victor et al.  Multitask prompted training enables zero-shot task generalization, 2021
> > >
> > > [3] Liang, Percy, et al. "Holistic Evaluation of Language Models." TMLR 2023.

---

> ### Author Response · Authors · 2024-06-04
> **Request for reconsideration of scores**
>
> Dear Reviewer, given that the reviewer-author discussion phase is coming to a close very soon, we'd greatly appreciate your input on whether there are any pending concerns that we can address which can help in reevaluation of your score.

---

> ### Author Response · Authors · 2024-06-07
>
> Dear Reviewer, given that the reviewer-author discussion phase is coming to a close very soon, we'd greatly appreciate your input on any further clarifications which can help in reevaluation of your score.

---

### Decision · Program_Chairs · 2024-07-10

**Decision:**

Accept

**Comment:**

This paper proposes a new template-based benchmark, building on several existing datasets, to measure biases related to gender and race in LLMs. The benchmark is then used in experiments that quantify bias in a range of LLMs.

The reviewers agree that the proposed benchmark would be a useful addition to current tools for quantifying the biases of LLMs. A few issues were raised regarding the methodology used to construct the dataset and how this may affect the results obtained, some of which were convincingly addressed during the discussion. Yet some concerns remain regarding comparison to other benchmarks, and the significance / interpretation of the experimental results.

[At least one review was discounted during the decision process due to quality]